# Accuracy of consensual stereotypes in moral foundations: A gender analysis

**Farhan Niazi, Ayesha Inam** * **, Zubaa Akhtar**

Department of Humanities, COMSATS University Islamabad, Tarlai Kalan, Islamabad, Pakistan

* Ayesha.inam@comsats.edu.pk

## Abstract

The current study explored the accuracy of consensual moral stereotypes that women and men hold about each other, as well as whether the gender differences in morality found in previous literature replicate on a sample of Pakistani individuals. A sample of 300 was used with an equal number of men and women. Data from 50 of the respondents was collected online, whereas the rest was collected in person from universities. The 30-item Moral Foundations Questionnaire (MFQ30) was used as a measure of five Moral Foundations, which are the basic elements of moral psychology as posited by Moral Foundations Theory. Men and women answered the questionnaire for themselves and then a second time, according to their perception of how a typical member of the other gender would respond which gave a measure of their stereotype. Comparison of actual scores of men and women revealed a statistically significant difference where women scored higher than men on the Harm foundation (p = 0.001). All other foundations, except for the Authority foundation showed the same pattern of differences as the previous literature, although they did not reach statistical significance. Stereotypes about men held by women were inaccurate underestimations on the Harm and Fairness foundations. The stereotype about women, held by men, was accurate on the Fairness foundation and inaccurate on the Authority foundation in the direction of underestimation. This research serves to further the study of Moral Foundations Theory as well as exploring the reasons behind the inaccurate moral stereotypes that men and women hold about each other, and actual gender differences in morality.

**Data Availability Statement:** All relevant data are within the manuscript and its Supporting Information files.

**Funding:** The author(s) received no specific funding for this work.

## Introduction

Psychological literature has generally considered inaccuracies and exaggerations as an inherent attribute of stereotypes [1] [2] [3]. However there are some serious problems with such a perspective, as mentioned in [4], because of which a general definition of stereotypes as "beliefs about the attributes of social groups" is proposed. This is in line with research that does not make inaccuracy a part of the definition on stereotypes [5]. A distinction can also be drawn between stereotypes held by a group and an individual: "Consensual stereotypes" are stereotypes shared by the members of a group, and are usually measured by means of samples, whereas, "personal stereotypes" are stereotypes held by an individual [6].

**Competing interests:** The authors have declared that no competing interests exist.

In keeping with this perspective, consensual stereotype accuracy (referred to as "stereotype accuracy" in the rest of this article) can be understood as the degree of correspondence between beliefs about groups and the actual characteristics of those groups [7]. This would involve assessing people's descriptive beliefs about groups, measuring the actual relevant characteristics of those groups (criteria), and then comparing beliefs to criteria. In this sense, stereotype accuracy has been noted to be one of the largest and most replicable effects in social psychology [6] [7]. Although, not all stereotypes are accurate, such as political stereotypes [5] [8] and national character stereotypes [9], there is considerable evidence for accuracy in many stereotypes [10] [11] [12].

Stereotypes about gender are not an exception to this pattern, as a review by Jussim in 2017 [4], of research on gender stereotypes, shows that most consensual stereotype judgments were accurate and most studies were dominated by accurate and near miss judgements with no evidence for the hypothesis that stereotypes generally cause exaggerations in real differences. In fact, underestimation of differences was also present [13] [14] [15]. Löckenhoff, et al., [16] found, from a sample of 3,323 individuals across 26 nations that consensual stereotypes about gender differences (perceived gender differences) on the Five Factor Model of personality mapped on well to actual assessed sex differences in personality.

Of the many types of stereotypes people can hold, one is that of "moral stereotypes" which can be understood as "beliefs about the morality of a social group". A review of the literature revealed only one publication in which Graham, Nosek, and Haidt [8], explored the consensual moral stereotypes that American liberals and conservatives have of each other. The researchers used the MFQ (Moral Foundations Questionnaire) that was constructed to measure five aspects or "foundations" of moral psychology (Harm, Fairness, In-group, Authority and Purity). They looked at how liberals and conservatives would rate endorsement of these five moral foundations for: themselves; a "typical" member of their own group and a "typical" member of the other group on the questionnaire. These ratings of a "typical" member of a group were called "moral stereotype" scores which were used to quantify moral stereotypes and subsequently consensual stereotypes. Consensual stereotypes, in this case, were found to be exaggerations of actual endorsement of moral foundations.

Evidence on moral stereotypes across genders is rare. A study [17] on women's implicit stereotypes revealed that, on average, men were considered to be more capable at violence whereas women were considered to be more competent, sociable and trustworthy than men. The authors note that no evidence could be found that these beliefs were harmful for women in terms of legitimizing their societal disadvantages, and that instead, they can be seen as serving the purpose of bolstering positive evaluations by women of their gender in-group. Another study [18] looked at how attractive women tend to be perceived as less truthful than their less attractive counterparts (dubbed the "femme fatale effect"). The results of six experiments suggested that, when delivering news of organizational change, this effect held for perceptions about women but not men. Furthermore, the effect persisted regardless of contextual changes such as whether the news being delivered was positive or negative, or whether the context was masculine or feminine. The researchers further argued that since this effect was eliminated when participants were primed to feel sexually secure, but not generally secure or sexually insecure, it could be understood as being rooted in sexual insecurity to some extent. Differing perspectives on gender also seem to come into play during evaluation of job candidates. One study [19] found that competence played a central role in the evaluation of both male and female job candidates, however, women were evaluated against even more criteria and their shortcomings were also more influential on their evaluations. The authors attribute this difference in evaluation strategies to the decision makers' belief that, compared to men, women would be more likely to encounter obstacles such as difficulties in being accepted and

respected in the work place (especially by men). Another possible explanation offered by the authors was that the evaluators might be concerned about the long term commitment of female candidates for their possible future decisions such as deciding to have a child.

Literature also indicates the presence of differences in morality across genders such as women being more likely to employ predominantly care oriented rather than justice-oriented considerations in moral dilemmas [20]. Women have also been noted to be more consistent in their use of care orientation and men have been noted more consistent in their use of justice orientation [21]. Graham, et al. [22], using their Moral foundations Questionnaire (MFQ), found that women, on average, scored greater than men on measures of care, fairness and purity; whereas men, on average, scored slightly greater than women on measures of authority and ingroup loyalty. A study [23] looking at gender differences in lying behavior found no clear evidence of differences in lying on an individual level; however on a group level more lying was noted in male and mixed groups than in female groups. Gender differences [24] have also been noted in ethical decision making where results of a meta-analysis revealed that, compared to men, women are more likely to consider specific business practices as unethical, however this difference declines as work experience increases.

Any discussion on morality must also involve cultural considerations because, in spite of there being certain commonalities in what is considered moral in multiple cultures such as honesty [25], it has been well documented how differences in culture are associated with differences in morality within and across societies [26]. One such example is of how Western, Educated, Industrialized, Rich, and Democratic (WEIRD) societies value individual autonomy and rights to a greater extent than non-WEIRD societies, which tend to prioritize community and spiritual purity [22]. There is no reason to assume that Pakistan is an exception to this rule and the moral profile of Pakistan should be studied with consideration to its cultural context. Applying Hofstede's [27][28] framework of cultural dimensions to Pakistani society indicates:

Pakistan's relatively high collectivist orientation, high propensity toward uncertainty avoidance, high power distance and masculinity largely account for many traditions and practices including strict adherence to hierarchy, centralization, corruption, nepotism and gender differentiation in administrative roles [29].

Indeed, gender segregation and male dominance can be noted in different mediums in Pakistan including public sector school textbooks [30], and has been a cause of diffuclty for women in the work place. The family structure is also typically hierarchical with strict roles defined in terms of age and gender, prioritizing the interrelatedness of family members as opposed to individual autonomy [31]. Religion being one of the strongest cultural influences on morality [32] must also not be ignored: the vast majority of Pakistanis are Muslims and Islamic ideology has been a central part of Pakistani constitution [33], leading to a conflict of views on gender rights and relations between traditionalists and institutions more sensitive to women's issues [34]. In keeping with these and a myriad of other cultural considerations, it would make sense to assume that the stereotypes held by Pakistanis, and their accuracy, would also be influenced by Pakistan's culture.

The morality literature has immensely focused on the gender differences with reference to moral judgments and how the two genders respond to moral dilemmas. The debate dates back to the end of the last century, with opposing views of Kohlberg & Gilligan about moral development and decision making in men and women [35]. According to Gilligan, women usually get stuck at earlier levels of this development, while men quite often move upward to abstract principles of morality. A lot of later empirical work has focused on investigating actual gender differences in moral decision-making and found differing results. While some found a statistically significant difference in moral decision-making between women and men [36], others did not [37]. A more recent meta-analysis conducted on 470 experimental studies indicated

that cheating and dishonest behavior is more prevalent in men than in women [38]. The literature also reveals that gender-related differences in moral judgment influence responses to dilemmas with emotionally salient actions. Men make pragmatic choices regardless of putting others at risk of danger or harm as compared to women [39].

These findings may be connected to the gender-differences in empathic ability [40][41] that make females more resistant to decisions that, despite being rationally viable, entail directly inflicting physical or moral pain to other individuals. Hence, female moral reasoning seems directed towards avoiding harming other people, and giving more priority to social relationships. Conversely, male moral thought hinges on the abstract principles of justice and fairness and on an individualistic stance [42].

The current research utilizes Moral Foundations Theory or "MFT" [43] as a framework from which to view morality. MFT tries to explain moral psychology by proposing multiple "basic" and "irreducible" foundations of moral thinking, grounded in evolutionary psychology, social psychology and anthropology. These "moral foundations" can be viewed as cognitive modules that detect relevant patterns in the environment and respond to them; with different people valuing these foundations to different extents, resulting in different moral systems and ideologies around the world.

Haidt and Joseph [44] identified five best candidates for these foundations: Care/Harm, which relates to recognizing signs of suffering and distress of people in general; Fairness/Cheating which relates to recognizing opportunities for mutual benefit and violations of agreements such as cheating in a game, or even improper functioning of inanimate objects such as a soda machine not working when you put in a dollar; Loyalty/Betrayal which relates to recognizing whether a person is acting in solidarity with or against a group they belong to; Authority/Subversion which relates to recognizing signs of obedience and deference in people; and Purity/Degradation which relates to recognizing signs of sacredness and impurity within a cultural context. It must be noted that these are by no means the only basic aspects of moral psychology, there may be more [22] but current research provides the most evidence in favor of these five foundations.

Exploring morality and related stereotypes across genders seems all the more important in light of recent movements against sexual harassment such as the "#MeToo" movement that quickly spread internationally through social media [45]. This movement gave greater impetus to the talk about sexual harassment and ethics [46] and even had an impact in Pakistan, with a sudden rise in allegations directed against prominent people [47]. Under such circumstances, it seems relevant to investiagte the beliefs about morality that men and women have of each other. Given the topic's contemporary relevance and the rare evidence on moral stereotypes and gender, the current research aims at examining this gap in the literature.

Moreover, as many aspects of morality do vary across cultures, the current study with its use of data from exclusively pakistani citizens, should serve to shed light on any culturally variable aspects of gender differences in moral foundations. It should also be a useful addition to the cross cultural examination of moral stereotypes as the results can be compared to any future research done on moral stereotypes across gender.

This research is primarily concerned with the accuracy of consensual stereotypes about morality, across genders i.e. the accuracy of stereotypes that women have about men and vice versa; with the population under consideration being Pakistani. It was also hypothesized that, as seen in Graham, et al. [22], women would show greater endorsement for Harm, Fairness and Purity foundations, whereas men wouldshow greater endorsement for Loyalty and Authority foundations.

The methodology is inspired by Graham, Nosek and Haidt [8] where the 30 item Moral Foundations Questionnaire was employed to study morality, and stereotypes about morality,

with respect to the five moral foundations. They used a seven point scale ranging from exremely conservative to exteremely liberal for people to identify their political affiliation. Participants were then asked to rate endorsement of the five moral foundations for: themselves; a "typical" member of their own group and a "typical" member of the other group on the questionnaire. The ratings of a "typical" member of a group were used to measure consensual moral stereotypes which were then compared to actual scores of extreme conservatives and extreme liberals. The reason for this was that scores from the extreme ends of the political spectrum would likely not be representative of the general population, so any stereotype scores equal to or greater than these would qualify as exaggerations. Scores from a nationally representative sample were also used as a benchmark of comparision.

The current study, utilizes a similar framework in its use of the Moral Foundation Questionnaire to measure morality and moral stereotypes, however, a nationally representative sample of the moral foundation scores of men and women was not available due to time constraints. It would also not make sense to measure the variable of gender on a scale with "extremes". Additionally, as it would already take time to fill the questionnaire once for onself and again for the other gender, providing additional scores for one's own gender would take too long and possibly lead to participant attrition. Therefore, inkeeping with the above considerations, it was decided that each participant would fill the questionnaire only twice to report two types of scores: actual scores for oneself, and stereotype scores about the other gender. The actual scores of each gender, aside from being compared to one another, would also be used as a criterion to which stereotype scores from the opposite gender would be compared for measuring accuracy.

## Methods

### Sample

A sample of 300 participants was collected, with an equal number of men and women. The men had a median age of 21 (IQR = 4) with a median of 14 (IQR = 3) years of education completed. The women also had a median age of 21 (IQR = 4) with a median of 15 (IQR = 2) years of education completed. Data from 50 of the respondents was from the online questionnaire whereas the rest was collected in person from universities.

The inclusion criteria restricted the sample to Pakistani citizens of age 18 and above, who identified as men or women, with a minimum of 12 years of education and ability to read, understand and write English. Purposive sampling was employed and data was collected both in person and through an online survey using google forms.

### Materials

The English version of the Moral Foundations Questionnaire by Graham, et al., [22] (referred to as MFQ30) was used, which is a 30 item, close-ended questionnaire that was last revised in July 2008 and is freely available at moralfoundations.org. The MFQ30 gives the mean score for endorsement of each foundation by an individual.

MFQ30 is composed of two parts, each with 15 items and 1 catch item. The first part called "moral relevance" is for measuring explicit thoughts about what is morally relevant with items rated on a 6 point likert scale from 0 (not at all relevant) to 5 (extremely relevant). The second part "moral judgement" is for assessing actual use of a moral foundation in judgement, which includes a 6 point likert scale from 0 (strongly disagree) to 5 (strongly agree).

Graham et al. [22] has previously reported alpha coefficients for each of the subscales as follows: Harm (0.69), Fairness (0.65), Ingroup (0.71), Authority (0.74) and Purity (0.84).

Since each participant completed two questionnaires, one of which they rated for themselves and the other for a typical member of the opposite gender, additions were made to the MFQ30 in terms of demographic questions asking about gender (men or women), age and years of education completed. Consensual stereotype held by a gender for each foundation was measured by averaging the predicted MFQ30 scores (about the opposite gender) on that foundation, from all the people of a gender. To gauge the accuracy of predictions about the opposite gender, we needed a standard of comparison. The most obvious comparison data were the actual ratings (average score for each foundation) provided by both genders in our sample, when they were asked to answer as themselves. Stereotype accuracy was measured for each foundation using discrepancy scores [6] of difference between consensual stereotype and a gender's actual score on that foundation.

In line with benchmarks proposed by Jussim [4], discrepancy scores at, and within, 10% or 0.25 standard deviations (SDs) of criteria considered accurate; scores more than 10% or 0.25 SDs off but at, or within, 20% or 0.50 SDs considered moderately accurate; and scores greater than 20% or 0.50 SDs inaccurate. Jussim [4] also mentions that these standards are useful for practical purposes but are ultimately arbitrary and a different standard can be used depending on certain purposes.

## Procedure

The current study was approved by Departmental Ethics Review Committee, Department of Humanities, COMSATS University Islamabad. The questionnaires were given to each participant along with an informed consent form, which can be found in S1 Appendix A. Each participant completed two sets of the MFQ. They filled out the standard MFQ for themselves and then the stereotype MFQ according to their perception of how a typical member of the opposite gender would respond. In doing so, they answered a total of 60 items.

Data was collected both in person and online. For online data collection, "google forms" was used to obtain data from participants who filled a duplicate of the questionnaires used in the study. To improve transparency of the data, all online participants had to sign in using their google email and were asked to leave their email as an optional choice. Written informed consent was taken to record their willingness.

## Analysis scheme

The syntax provided at moralfoundations.org was used for computing mean score of a foundation for each person, which was then used to calculate the mean score of all people of a gender. A bootstrapped Welch's t-test [48] was used for testing the hypothesis about differences in group mean in the population. Welch's t-test was preferred over the Student's t-test as it performs better in the context of psychological research [49], specifically with regard to the violation of the homogeneity of variance assumption.

15 tests (3 per foundation) were done to study group differences in order to make inferences about the target population. For each foundation:

1. a significance test was done to calculate difference in "endorsement by gender"

2. a significance test was done to calculate the difference between "endorsement by gender" of men and "consensual stereotype" about men

3. a significance test was done to calculate the difference between "endorsement by gender" of women and "consensual stereotype" about women

**Table 1. Descriptive statistics and alpha coefficients of each subscale (N = 300).**

| Variable | No. of Items | *M* | SD | A | Range | | Skew |
|---|---|---|---|---|---|---|---|
| | | | | | Potential | Actual | |
| MFQ30 | | | | | | | |
| Harm Subscale | 6 | 3.91 | 0.69 | .53 | 0–5 | 0.83–5 | -0.95 |
| Fairness Subscale | 6 | 3.92 | 0.58 | .43 | 0–5 | 1.83–5 | -.066 |
| Ingroup subscale | 6 | 3.24 | 0.79 | .50 | 0–5 | 1–5 | -0.62 |
| Authority Subscale | 6 | 3.40 | 0.63 | .41 | 0–5 | 1.17–4.67 | -0.62 |
| Purity Subscale | 6 | 3.57 | 0.75 | .58 | 0–5 | 1–5 | -0.70 |

MFQ30 = Moral Foundations Questionnaire (30 item version). Mean, Standard Deviation Ranges and skew reported for aggregate (mean) score of each person on the scale.

The alpha level chosen was 0.05. However, since multiple tests on the same data inflate type-1 error [50], the Bonferroni correction was used to control for family wise error rate. This was done by dividing the alpha by number of comparisons to give the adjusted alpha. For the current study, this was the alpha (0.05) divided by number of comparisons (15), which gives an adjusted alpha of 0.00333... that was rounded to 0.003 for use in significance tests.

## Results

Table 1 shows the alpha co-efficient and skewness of all subscales comprising the Moral Foundations Questionnaire (30 item version). The table shows low to medium internal consistency for all scales.

Analysis revealed statistically significant differences at the corrected alpha level. Women scored higher than men on the Care foundation, and the stereotypes they held about men were inaccurate, with men scoring higher than the stereotypes about them. Women also scored higher than the stereotypes about them, which were accurate, though this difference was not statistically significant (Table 2).

The results revealed statistically significant differences at the corrected alpha level. The stereotypes that women held about men were inaccurate with men scoring higher than the stereotypes. The stereotypes that men held about women were moderately accurate with women scoring lower than the stereotypes. Although women scored higher than men on the Fairness subscale, this difference was not statistically significant (Table 3).

No statistically significant differences were found at the corrected alpha level for comparison of gender on the Ingroup subtest. Men scored slightly higher than women on the Ingroup foundation, and the stereotypes that women held about men were moderately accurate with

**Table 2. Comparison of gender on actual and stereotype scores on harm subscale (N = 300).**

| | Men (*n* = 150) | Women (*n* = 150) | | 95% Bootstrapped CI | | Cohen's *ds* |
|---|---|---|---|---|---|---|
| Harm Subscale | *M(SD)* | *M(SD)* | *t*(298) | LL | UL | |
| Actual Scores | 3.71(0.76) | 4.11(0.54) | 5.25* | 0.26 | 0.53 | 0.61 |
| MA vs SAM | 3.71(0.76) | 3.16(0.96) | 5.57* | 0.37 | 0.75 | 0.64 |
| WA vs SAW | 4.03(0.77) | 4.11(0.54) | 1.08 | - 0.05 | 0.23 | 0.13 |

MA = Men Actual, SAM = Stereotype About Men, WA = Women Actual, SAW = Stereotype About Women,

CI = Confidence Interval; LL = Lower Limit; UL = Upper Limit

*p< .003

**Table 3. Comparison of gender on actual and stereotype scores on fairness subscale (N = 300).**

| Fairness Subscale | Men (*n* = 150) M(SD) | Women (*n* = 150) M(SD) | *t*(298) | 95% Bootstrapped Cl LL | UL | Cohen's *ds* |
|---|---|---|---|---|---|---|
| Actual Scores | 3.82(0.61) | 4.01(0.54) | 2.89 | 0.08 | 0.32 | 0.33 |
| MA vs SAM | 3.82(0.61) | 3.42(0.78) | 4.92* | 0.24 | 0.56 | 0.57 |
| WA vs SAW | 4.01(0.54) | 3.77(0.71) | 3.31* | 0.10 | 0.39 | 0.38 |

MA = Men Actual, SAM = Stereotype About Men, WA = Women Actual, SAW = Stereotype About Women,

CI = Confidence Interval; LL = Lower Limit; UL = Upper Limit

*p< .003

men scoring lower than the stereotypes. The stereotypes that men held about women were accurate with women scoring lower than the stereotypes (Table 4).

Only one statistically significant difference at the corrected alpha level was revealed. The stereotypes that men held about women were inaccurate just above moderate accuracy with women scoring higher than the stereotypes. Women scored barely higher than men on the Authority subscale however this difference was not statistically significant. The stereotypes that women held about men were also accurate with men scoring slightly lower than the stereotypes, although this difference was also not statistically significant (Table 5).

No statistically significant differences were found at the corrected alpha level for comparison of gender on the Purity subscale. Women scored higher than men, and the stereotype they held about men were accurate with men scoring higher than the stereotype. The stereotype about women held by men was also accurate with women scoring slightly less than the stereotype (Table 6).

## Discussion

The current study aimed to assess the accuracy of moral stereotypes that men and women hold about each other. This was done using the MFQ30 as a measure of morality in the form of endorsement of moral foundations. Men and women in the sample filled the questionnaire according to their own beliefs and also according to how they thought a typical member of the opposite gender would respond, which would reveal the stereotypes they held. The stereotypes about each gender were then compared to the actual responses given by them, to measure accuracy. The actual scores of men and women were also compared to each other in order to replicate the results from previous research.

**Table 4. Comparison of gender on actual and stereotype scores on ingroup subscale (N = 300).**

| Ingroup Subscale | Men (*n* = 150) M(SD) | Women (*n* = 150) M(SD) | *t*(298) | 95% Bootstrapped Cl LL | UL | Cohen's *ds* |
|---|---|---|---|---|---|---|
| Actual Scores | 3.31(0.79) | 3.17(0.78) | 1.48 | - 0.05 | 0.32 | 0.17 |
| MA vs SAM | 3.31(0.79) | 3.51(0.77) | 2.22 | 0.02 | 0.38 | 0.26 |
| WA vs SAW | 3.33(0.72) | 3.17(0.78) | 1.83 | - 0.01 | 0.32 | 0.21 |

MA = Men Actual, SAM = Stereotype About Men, WA = Women Actual, SAW = Stereotype About Women,

CI = Confidence Interval; LL = Lower Limit; UL = Upper Limit

*p< .003

**Table 5. Comparison of gender on actual and stereotype scores on authority subscale (N = 300).**

| Authority Subscale | Men (*n* = 150) M(SD) | Women (*n* = 150) M(SD) | *t*(298) | 95% Bootstrapped CI LL | UL | Cohen's *ds* |
|---|---|---|---|---|---|---|
| Actual Scores | 3.37(0.65) | 3.44(0.61) | 0.92 | - 0.07 | 0.21 | 0.1 |
| MA vs SAM | 3.37(0.65) | 3.43(0.81) | 0.68 | - 0.1 | 0.23 | 0.08 |
| WA vs SAW | 3.08(0.78) | 3.44(0.61) | 4.45* | 0.21 | 0.50 | 0.51 |

MA = Men Actual, SAM = Stereotype About Men, WA = Women Actual, SAW = Stereotype About Women,

CI = Confidence Interval; LL = Lower Limit; UL = Upper Limit

*p< .003

MFQ30 showed low to moderate levels of alpha reliability in the current study. With the exception of the Sanctity subscale, each subscale showed negative inter-item correlations, with some subscales showing improved alpha reliability upon deletion of certain items. For example, on the Care subscale, the item "it can never be right to kill a human being" negatively correlated with the items "someone was cruel" and "compassion for those who are suffering is the most crucial virtue". This makes sense in terms of the cultural and religious context of Pakistan, where people might believe that there are cases where killing another person is justified, while at the same time considering cruelty and compassion to be very relevant to their thinking about morality. Similarly, on the Fairness foundation, the item "I think it's morally wrong that rich children inherit a lot of money while poor children inherit nothing" negatively correlated with "someone acted unfairly" and "someone was denied his or her rights". This could be due to a culture of strong kinship ties and inheritance of wealth, along with strict adherence to hierarchy [29], where people would* getting large amounts of inheritance, and concentration of wealth with a few people in authoritative postitions tocould be seen as being justified. This could be the case even though people do care about fairness in general. Indeed it must be noted that cultures do vary in terms of which aspects of the same moral dimension they give preference to [26]. On the Ingroup scale, the item "someone did something to betray his or her group" negatively correlated with "I am proud of my country's history". This might be due to the fact that Pakistan is culturally diverse with many different communities and ethnicities [51], which may feel more affiliation towards each other than the country as a whole. On the Authority subscale, the item "An action caused chaos and disorder" negatively correlated with "Men and women each have different roles to play in society" and "If I were a soldier and disagreed with my commanding officer's orders, I would obey anyway because that is my duty". The reasons for such a negative correlation are not clear. Overall, the pattern of negative

**Table 6. Comparison of gender on actual and stereotype scores on purity subscale (N = 300).**

| Purity Subscale | Men (*n* = 150) M(SD) | Women (*n* = 150) M(SD) | *t*(298) | 95% Bootstrapped CI LL | UL | Cohen's *ds* |
|---|---|---|---|---|---|---|
| Actual Scores | 3.47(0.73) | 3.67(0.76) | 2.35 | 0.02 | 0.37 | 0.27 |
| MA vs SAM | 3.47(0.73) | 3.26(0.92) | 2.17 | 0.03 | 0.4 | 0.25 |
| WA vs SAW | 3.73(0.68) | 3.67(0.76) | 0.68 | -0.11 | 0.22 | 0.08 |

MA = Men Actual, SAM = Stereotype About Men, WA = Women Actual, SAW = Stereotype About Women,

CI = Confidence Interval; LL = Lower Limit; UL = Upper Limit

*p< .003

correlations suggests that differences between the culture in which MFQ30 was developed and the Pakistani culture, may have contributed to the low reliability and might indicate that validity was affected too as some items of the same subscale were not correlated.

Comparison of actual scores of men and women on the MFQ30 showed a statistically significant difference where women scored higher than men on the Harm foundation. Specifically, in the context of Pakistani culture, this can be explained by how Pakistan scores higher on "masculinity" [29] on Hofstede's [27][28] cultural dimensions, characteristics of which include assertiveness and competitiveness, as opposed to femininity which involves caring for others and being more cooperative. All other differences were not statistically significant, and it remains to be seen whether with a larger sample size these differences turn out to be statistically significant.

With regard to the direction of the differences, all except one foundation showed the same pattern of difference as in [22]. As before, women scored higher than men on Care, Fairness and Sanctity, whereas men scored higher on Ingroup. However, as opposed to the findings of the previous research, women scored higher on Authority as well, with a greater effect size than was seen in Graham, et al. [22]. where men scored barely higher than women. One explanation for this result could be the internalization of patriarchy by women in Pakistani society [52], resulting in greater subordination and deference to authority. Overall, the direction of differences suggests that gender differences in moral foundation endorsement, follows a similar pattern in Pakistan as in the large international sample from Graham, et al. [22].

Previous research on consensual gender stereotypes has shown them to be generally accurate. For example, McCauley & Thangavelu [13] reported that stereotypes about the proportion of men and women in different occupations were predominantly accurate. Swim [14] also reported that stereotypes about sex differences among various characteristics, tended towards accuracy. Similarly, Beyer [15] found that the stereotypes about the percentage of women and men students in different academic fields as well as their GPAs were mostly accurate. Predominance of accuracy among stereotypes was also reported by Diekman, Eagly, and Kulesa [53] where attitudes of men and women regarding social and political issues were also found to range from accuracy for consensual discrepancy scores to near misses for personal discrepancies. Löckenhoff, et al. [16] also reported that consensual stereotypes about gender differences (perceived gender differences) on the Five Factor Model of personality were considerably accurate.

The current study however shows a different pattern, at least among the statistically significant results where most stereotypes were inaccurate. Two stereotypes held by women were inaccurate underestimations (Harm, Fairness), whereas one of the stereotypes held by men was an accurate overestimation (fairness) and one was a barely inaccurate underestimation (authority). This shows that overall; men were more accurate than women. For the other foundations, no inferences can be made due to not having reached statistical significance.

This suggests that women in Pakistan with similar characteristics as the sample in the current study might underestimate the importance men give to caring for, and dealing fairly with others. This could be reflective of how patriarchal [54] oppression of women in Pakistan [55] [56] might have resulted in them developing expectations of an uncaring and unjust attitude from men. These expectations however do not match up with data from the men in the sample, possibly because of differences between men that attend university and those who do not. Indeed, as noted by Graham et al [26], people from WEIRD (Western, Educated, Industrialized, Rich and Democratic) segments of society hold different moral values than those from non-WEIRD backgrounds, and the men in this study might be different from those that do not go to university but still have an influence on the stereotypes held by women.

Men also underestimated on the Authority foundation and were inaccurate. This effect should also be explored further with attempts at replication.

It can be concluded that only one statistically significant difference between men and women could be found, which was on the Harm foundation. However, the direction of gender differences was the same as in Graham, et al. [22] for all foundations with the exception of one: Authority, where women scored higher than men. If these results indeed replicate and show a statistically significant difference with a larger sample, this might indicate that patterns of moral foundation endorsement related to gender may be similar in Pakistan as in other cultures, with some differences.

Future research should aim explore the determinants of over and under estimation in stereotypes, as well as why women underestimated the endorsement of Harm and Fairness by men and why men underestimated on Authority. This study should serve as a launch pad for further work on both moral psychology and the study of the Moral Foundations framework in the indigenous setting. Patterns of gender differences in the endorsement of multiple Moral Foundations should help in policy making and communication within organizations and institutions involving both men and women in order to better facilitate both genders.

## Limitations and recommendations

Some limitations of the current study must be acknowledged. The sampling strategy was non-random and consisted mostly of university students, which would limit the generalizability of the results. The standard of comparison to test the stereotypes against was also not representative of the Pakistani population as it was taken from the sample itself due to resource limitations. We recommend that generalization of these results is restricted to university students with similar age and education levels as in the sample.

Future research should aim to validate the Urdu version of the MFQ30 should on a representative sample to verify the factor structure as well as the reliability of the scale in Pakistan in order to explore the reasons for low alpha reliability and negative inter-item correlations in the current study.

Further studies should also try to replicate the results of this study on a larger and more representative probability sample, and explore the determinants of over and under estimation in stereotypes, as well as why females underestimated the endorsement of Care and Fairness by males, why males underestimated on Authority.

This study was the first of its kind in utilizing the Moral Foundations framework and the MFQ30 on a sample consisting solely of Pakistanis, through both in person and online data collection. As such it should serve as a launch pad for further work on both moral psychology and the study of the Moral Foundations framework in the Pakistani context.

Patterns of gender differences in the endorsement of multiple Moral Foundations should help in policy making and communication within organizations and institutions involving both males and females in order to better facilitate both genders. The general accuracy of gender stereotypes regarding moral foundations should also serve to indicate that males' and females' perception of each other's morality is mostly accuarate to moderately accurate. However, females with similar characteristics to the sample in this study might underestimate the endorsement of Care and Fairness foundations by males in general and this should serve to make the general public look into the matter as it could be the result of distorted perceptions from the media among other reasons. The same holds true for underestimation of females' endorsement of Authority by males in the sample.

## Supporting information

**S1 Appendix.**
(DOCX)

**S1 File.**
(SAV)

## Acknowledgments

The author acknowledges COMSATS University Islamabad for providing platform to conduct this research.

## Author Contributions

**Conceptualization:** Farhan Niazi, Ayesha Inam.

**Data curation:** Farhan Niazi.

**Formal analysis:** Farhan Niazi, Ayesha Inam.

**Investigation:** Farhan Niazi.

**Methodology:** Farhan Niazi.

**Supervision:** Ayesha Inam.

**Writing – original draft:** Farhan Niazi.

**Writing – review & editing:** Farhan Niazi, Ayesha Inam, Zubaa Akhtar.

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
