## [Decision Letter · Decision Letter 0]

4 Nov 2019

PONE-D-19-25204

Title: Accuracy of Consensual Stereotypes in Moral Foundations: A Gender Analysis

PLOS ONE

Dear Dr. Inam,

Thank you for submitting your manuscript to PLOS ONE. After careful consideration, we feel that it has merit but does not fully meet PLOS ONE’s publication criteria as it currently stands. Therefore, we invite you to submit a revised version of the manuscript that addresses the points raised during the review process.

Please find the reviewers' comments below, as well as those from my own.

We would appreciate receiving your revised manuscript by Dec 19 2019 11:59PM. To enhance the reproducibility of your results, we recommend that if applicable you deposit your laboratory protocols in protocols.io, where a protocol can be assigned its own identifier (DOI) such that it can be cited independently in the future. For instructions see: http://journals.plos.org/plosone/s/submission-guidelines#loc-laboratory-protocols

We look forward to receiving your revised manuscript.

Kind regards,

Valerio Capraro

Academic Editor

PLOS ONE

Journal Requirements:

2. Thank you for your ethics statement: "Written informed consent was taken to record their willingness. Informed consent was provided by the departmental ethics review committee, which was developed according to ethical standards of American Psychological Association."

Additional Editor Comments:

I have now collected two reviews from two experts in the field. The reviewers like the paper but suggest several improvements. Therefore, I would like to invite you to revise your manuscript according to their comments. Additionally, I would like to add one more comment regarding the literature review, which ignores the enormous literature on gender differences in various domains of morality. For example: gender differences in moral judgments in moral dilemmas (Fumagalli et al. 2010; Friesdorf et al. 2015; Capraro & Sippel, 2017), gender differences in honesty (Capraro, 2018; Abeler et al. 2019; Gerlach et al. 2019); gender differences in cooperation (Rand, 2017); gender differences in altruism (Rand et al. 2016; Branas-Garza et al. 2018). Of course it is not a requirement to cite these specific works, but I think that, anyway, the literature review on gender differences should be highly improved.

I am looking forward for the revision.

References

J Abeler, D Nosenzo, C Raymond (2019) Preferences for truth‐telling. Econometrica 87 (4), 1115-1153

P Brañas-Garza, V Capraro, E Rascón-Ramírez (2018) Gender differences in altruism on mechanical turk: Expectations and actual behaviour. Economics Letters 170, 19-23

V Capraro (2018) Gender differences in lying in sender-receiver games: A meta-analysis. Judgment and Decision Making 13, 345-355

V Capraro, J Sippel (2017) Gender differences in moral judgment and the evaluation of gender-specified moral agents. Cognitive processing 18 (4), 399-405

Friesdorf, R., Conway, P., & Gawronski, B. (2015). Gender differences in responses to moral dilemmas: a process dissociation analysis. Personality and Social Psychology Bulletin, 41(5), 696-713.

Fumagalli, M., Ferrucci, R., Mameli, F., Marceglia, S., Mrakic-Sposta, S., Zago, S., ... & Cappa, S. (2010). Gender-related differences in moral judgments. Cognitive processing, 11(3), 219-226.

Gerlach, P., Teodorescu, K., & Hertwig, R. (2019). The truth about lies: A meta-analysis on dishonest behavior. Psychological bulletin, 145(1), 1.

DG Rand, VL Brescoll, JAC Everett, V Capraro, H Barcelo (2016) Social heuristics and social roles: Intuition favors altruism for women but not for men. Journal of Experimental Psychology: General 145 (4), 389-396

Reviewers' comments:

Reviewer's Responses to Questions

**Comments to the Author**

1. Is the manuscript technically sound, and do the data support the conclusions?

Reviewer #1: Yes

Reviewer #2: Partly

2. Has the statistical analysis been performed appropriately and rigorously? 

Reviewer #1: Yes

Reviewer #2: Yes

3. Have the authors made all data underlying the findings in their manuscript fully available?

Reviewer #1: Yes

Reviewer #2: No

4. Is the manuscript presented in an intelligible fashion and written in standard English?

Reviewer #1: Yes

Reviewer #2: No

5. Review Comments to the Author

Reviewer #1: Summary

The contribution investigated in a 300 Pakistani adults the moral stereotypes that males and females hold about each other and the influences of gender in such moral stereotypical representations using the Moral Foundation Questionnaire (MFQ30). Participants were asked in two times to respond (1) for themselves (actual score) and (2) how a typical member of the other gender group, according to their perception, would answer (other gender stereotypes score). The Females scored higher than Males only on the Care Foundation sub-scale. Females underestimated Stereotypes about Males on the Harm and Fairness foundations; Males were accurate on Fairness foundation of Females, whereas underestimated their Authority foundation.

Comments

In general the topic of the consensual stereotypes about moral foundation attributed to the other gender group, is relevant and not yet studied. The theoretical frame is pertinent and the relevance of the study is also well motivated by the recent spread of anti-sexist movements not only in the western world. Nevertheless I was surprised by the absence of any mention to the debate about the cross-cultural differences in the people morality judgement (e.g. Graham et al., 2011 for East- West differences in foundation endorsement, even if as the effect sizes in their results showed, the gender differences were much stronger than the differences between Eastern and Western cultures).

It could be necessary for introducing this study to propose some considerations and references about the main characteristics of Pakistani culture which certainly influences the relationships between genders and may partly explain possible differences across gender stereotypes. In fact expectations about the specificity of cultural context of participants are missing, nor this relevant aspect has been sufficiently covered in the discussion. At last in the Introduction Section would be useful to refer to empathy disposition, as a recognized discriminant factor between several cross gender dimensions.

.

Moreover I have concerns about some aspects of the methodological approach.

1. The choice of the sample, constituted in prevalence by university students, is not representative of the Pakistani population. This must be acknowledged among the limitations of the study;

2. Alpha coefficients for single scale in the sample examined are missing;

Minor points:

1. Why in all tables the average scores (Actual and Stereotypes scores) of Males (MA vs, SAM) and Females (SAF vs. FA) are reported in a different order? This is confusing.

2. The labels of the 5 moral Foundations are differently used in the different parties of the text. For example the Harm/Care sub-scale in prevalence is called simply Care sub-scale, sometimes it is called Harm sub-scale. On the other side, Graham et al. in their validation study of MFQ (2011) called the 5 sub-scales using the first label for each couple (Harm, Fairness, In-group, Authority, Purity). For more clarity and consistency in the interpretation of the results obtained, according to my opinion it would be better using always the complete labels of the 5 moral bi-dimensional Foundations (Harm/Care, Fairness/Reciprocity, In-group/Loyalty, Authority/Respect, Purity/Sanctity).

Reviewer #2: In the manuscript the authors present one study aimed at examining consensual moral stereotypes that women and men hold about each other. The topic is innovative and contributes to the literature on social judgment and morality. It is also interesting to read findings about the Pakistani population since to my knowledge, most studies on morality concern populations from Western countries. I have however some concerns, regarding, in particular, how the study is presented at the conceptual level.

Indeed, I do not understand why the study is presented as a research on stereotype accuracy. According to Jussim et al. (2015), who are cited in the manuscript, the assessment of stereotype accuracy implies a comparison between people’s beliefs about a group and some criteria that establish group characteristics (i.e., more objective data such Census data about the proportion of people who have certain characteristics or perform specific behaviors). In the present manuscript, such a comparison is not done. If I understood well, the authors claim that a “standard of comparison” can be represented by ratings of women and men about themselves. However, it seems to me that these can be intended as self-construal ratings rather than criteria that allow to grasp stereotype accuracy. This is my major concern. I think a different framework for the research would increase the impact and the readability of the paper.

I would also stress the importance of collecting such data in the Pakistani context, and would discuss in a deeper way how these results can be specific for that context (or can be generalized to others). This argument might also be developed in the future directions section.

With respect to the implications of the research, the authors anticipate in the introduction the importance of exploring morality and related stereotypes in the light of the #MeToo movement. I think a similar reasoning should be moved to (and expanded in) the discussion.

Moreover, I find that the very last paragraph of the discussion should be expanded. The authors should provide more reasoning on the practical implications of this study.

Minor points:

I find it very hard to follow the section results due to the numerous acronyms used in the Tables (MA, SAM, SAF, FA)

I am not an English-native speaker, but I have the impression that the manuscript would benefit from a linguistic revision. Moreover, to my knowledge the APA style does not use male and female as nouns, but only as adjectives. I would rather use women and men as nouns.

6. PLOS authors have the option to publish the peer review history of their article (what does this mean?). If published, this will include your full peer review and any attached files.

Reviewer #1: Yes: Carmen Belacchi

Reviewer #2: No

---

## [Author Response · Author response to Decision Letter 0]

7 Jan 2020

Rebuttal Letter

Comments

1. Is the manuscript technically sound, and do the data support the conclusions?

Reviewer #1: Yes

Reviewer #2: Partly

No response required.

2. Has the statistical analysis been performed appropriately and rigorously? 

Reviewer #1: Yes

Reviewer #2: Yes

 No response required.

3. Have the authors made all data underlying the findings in their manuscript fully available?

Reviewer #1: Yes

Reviewer #2: No

Data is provided as supporting file.

4. Is the manuscript presented in an intelligible fashion and written in standard English?

Reviewer #1: Yes

Reviewer #2: No

Corrected after rigorous proof-reading. 

 

Reviews 

Editor’s Comments 

1.When submitting your revision, we need you to address these additional requirements. Please ensure that your manuscript meets PLOS ONE's style requirements, including those for file naming. The PLOS ONE style templates can be found at http://www.journals.plos.org/plosone/s/file?id=wjVg/PLOSOne_formatting_sample_main_body.pdf and http://www.journals.plos.org/plosone/s/file?id=ba62/PLOSOne_formatting_sample_title_authors_affiliations.pdf

Edited accordingly

2. Thank you for your ethics statement: "Written informed consent was taken to record their willingness. Informed consent was provided by the departmental ethics review committee, which was developed according to ethical standards of American Psychological Association."

Added in ethics statement (Line 235- 236).

“b) If there are no restrictions, please upload the minimal anonymized data set necessary to replicate your study findings as either Supporting Information files or to a stable, public repository and provide us with the relevant URLs, DOIs, or accession numbers.” is the choice suitable. 

Data is uploaded as supporting file.

Additionally, I would like to add one more comment regarding the literature review, which ignores the enormous literature on gender differences in various domains of morality. For example: gender differences in moral judgments in moral dilemmas (Fumagalli et al. 2010; Friesdorf et al. 2015; Capraro & Sippel, 2017), gender differences in honesty (Capraro, 2018; Abeler et al. 2019; Gerlach et al. 2019); gender differences in cooperation (Rand, 2017); gender differences in altruism (Rand et al. 2016; Branas-Garza et al. 2018). Of course, it is not a requirement to cite these specific works, but I think that, anyway, the literature review on gender differences should be highly improved.

I am looking forward for the revision.

Added in the literature review (Line 109- 127)

Editor’s Comments Received on December 23rd, 2019 

In your ethics statement you write:

"Written informed consent was taken to record their willingness. Informed consent was provided by the Departmental Ethics Review Committee, Department of Humanities, COMSATS University Islamabad, which was developed according to ethical standards of American Psychological Association."

Please specify whether the Departmental Ethics Review Committee, Department of Humanities, COMSATS University Islamabad approved the current study.

The current study was approved by Departmental Ethics Review Committee, Department of Humanities, COMSATS University Islamabad (Line 235- 236). 

Editor’s Comments Received on January 2nd, 2020 

Thank you for the additional edits. For clarity and to avoid confusion, we recommend removing the statement reading, "Informed consent was provided by the Departmental Ethics Review Committee, Department of Humanities, COMSATS University Islamabad, which was developed according to ethical standards of American Psychological Association."

Also, in the manuscript, you note a consent form is provided in Appendix A, but we cannot find a consent form or an appendix. Please update this before we proceed.

Removed as directed. Attached informed consent in the end as Appendix A. 

Reviewer 1

Nevertheless I was surprised by the absence of any mention to the debate about the cross-cultural differences in the people morality judgement (e.g. Graham et al., 2011 for East- West differences in foundation endorsement, even if as the effect sizes in their results showed, the gender differences were much stronger than the differences between Eastern and Western cultures).

It could be necessary for introducing this study to propose some considerations and references about the main characteristics of Pakistani culture which certainly influences the relationships between genders and may partly explain possible differences across gender stereotypes. In fact expectations about the specificity of cultural context of participants are missing, nor this relevant aspect has been sufficiently covered in the discussion. At last in the Introduction Section would be useful to refer to empathy disposition, as a recognized discriminant factor between several cross-gender dimensions.

Added in introduction (Line 84-108 & Line 155- 159).

Added in discussion (Lines 361- 364, 398- 405). 

Moreover, I have concerns about some aspects of the methodological approach.

1. The choice of the sample, constituted in prevalence by university students, is not representative of the Pakistani population. This must be acknowledged among the limitations of the study;

2. Alpha coefficients for single scale in the sample examined are missing;

1. Added in discussion (Lines 422- 427).

2. Added in results (Line 266- 273) & Added in discussion (Lines: 331-358).

Minor points:

1. Why in all tables the average scores (Actual and Stereotypes scores) of Males (MA vs, SAM) and Females (SAF vs. FA) are reported in a different order? This is confusing.

2. The labels of the 5 moral Foundations are differently used in the different parties of the text. For example the Harm/Care sub-scale in prevalence is called simply Care sub-scale, sometimes it is called Harm sub-scale. On the other side, Graham et al. in their validation study of MFQ (2011) called the 5 sub-scales using the first label for each couple (Harm, Fairness, In-group, Authority, Purity). For more clarity and consistency in the interpretation of the results obtained, according to my opinion it would be better using always the complete labels of the 5 moral bi-dimensional Foundations (Harm/Care, Fairness/Reciprocity, In-group/Loyalty, Authority/Respect, Purity/Sanctity).

1.Corrected in all tables as “MA = Men Actual, SAM = Stereotype About Men, WA = Women Actual, SAW = Stereotype About Women.”

2. Corrected throughout text as “Harm, Fairness, In-group, Authority, Purity” according to Graham et al. (2011).

Reviewer 2 

I have however some concerns, regarding, in particular, how the study is presented at the conceptual level.

Indeed, I do not understand why the study is presented as a research on stereotype accuracy. According to Jussim et al. (2015), who are cited in the manuscript, the assessment of stereotype accuracy implies a comparison between people’s beliefs about a group and some criteria that establish group characteristics (i.e., more objective data such Census data about the proportion of people who have certain characteristics or perform specific behaviors). In the present manuscript, such a comparison is not done. If I understood well, the authors claim that a “standard of comparison” can be represented by ratings of women and men about themselves. However, it seems to me that these can be intended as self-construal ratings rather than criteria that allow to grasp stereotype accuracy. This is my major concern. I think a different framework for the research would increase the impact and the readability of the paper.

Added in Introduction (Line 168-192)

I would also stress the importance of collecting such data in the Pakistani context, and would discuss in a deeper way how these results can be specific for that context (or can be generalized to others). This argument might also be developed in the future directions section.

Pakistani culture and other aspects relevant to the context of the research have now been mentioned in the introduction and discussion section. (Line 84-108, 155- 159, 361- 364, 398- 405)

With respect to the implications of the research, the authors anticipate in the introduction the importance of exploring morality and related stereotypes in the light of the #MeToo movement. I think a similar reasoning should be moved to (and expanded in) the discussion. Moreover, I find that the very last paragraph of the discussion should be expanded. The authors should provide more reasoning on the practical implications of this study.

Added in discussion (Line 398- 405)

Minor points:

I find it very hard to follow the section results due to the numerous acronyms used in the Tables (MA, SAM, SAF, FA)

I am not an English-native speaker, but I have the impression that the manuscript would benefit from a linguistic revision. Moreover, to my knowledge the APA style does not use male and female as nouns, but only as adjectives. I would rather use women and men as nouns.

1. Corrected in all tables as “MA = Men Actual, SAM = Stereotype About Men, WA = Women Actual, SAW = Stereotype About Women.”

2. Corrected as Men & Women across paper.

---

## [Decision Letter · Decision Letter 1]

28 Jan 2020

PONE-D-19-25204R1

Title: Accuracy of Consensual Stereotypes in Moral Foundations: A Gender Analysis

PLOS ONE

Dear Dr. Inam,

Thank you for submitting your manuscript to PLOS ONE. After careful consideration, we feel that it has merit but does not fully meet PLOS ONE’s publication criteria as it currently stands. Therefore, we invite you to submit a revised version of the manuscript that addresses the points raised during the review process.

We would appreciate receiving your revised manuscript by Mar 13 2020 11:59PM. To enhance the reproducibility of your results, we recommend that if applicable you deposit your laboratory protocols in protocols.io, where a protocol can be assigned its own identifier (DOI) such that it can be cited independently in the future. For instructions see: http://journals.plos.org/plosone/s/submission-guidelines#loc-laboratory-protocols

We look forward to receiving your revised manuscript.

Kind regards,

Valerio Capraro

Academic Editor

PLOS ONE

Additional Editor Comments (if provided):

One reviewer still has some minor suggestions. Please address them at your earliest convenience. I am looking forward for the final version.

Reviewers' comments:

Reviewer's Responses to Questions

**Comments to the Author**

1. If the authors have adequately addressed your comments raised in a previous round of review and you feel that this manuscript is now acceptable for publication, you may indicate that here to bypass the “Comments to the Author” section, enter your conflict of interest statement in the “Confidential to Editor” section, and submit your "Accept" recommendation.

Reviewer #1: All comments have been addressed

Reviewer #2: All comments have been addressed

2. Is the manuscript technically sound, and do the data support the conclusions?

Reviewer #1: Yes

Reviewer #2: Yes

3. Has the statistical analysis been performed appropriately and rigorously? 

Reviewer #1: Yes

Reviewer #2: Yes

4. Have the authors made all data underlying the findings in their manuscript fully available?

Reviewer #1: Yes

Reviewer #2: Yes

5. Is the manuscript presented in an intelligible fashion and written in standard English?

Reviewer #1: Yes

Reviewer #2: Yes

6. Review Comments to the Author

Reviewer #1: (No Response)

Reviewer #2: In the revised version, the authors have addressed the points raised by me and the other reviewer. In particular, I appreciated the adding of more considerations and references to the Pakistani context, which help understand and contextualize the findings.

I only have few concerns, which are outlined below.

With respect to gender differences in morality, the authors mostly reported studies on moral judgment. Even though I agree that there has been little research on morality in gender stereotypes, I suggest that the authors look at and eventually add the evidence of Leach, Carraro, Garcia, & Kang (2017) on stereotype of women as more trustworthy than men; Sheppard and Johnson (2019) on attractiveness and trustworthiness; Moscatelli, Menegatti et al. (2020, advanced online publication on Sex Roles) on the importance of morality for women in employment evaluation. Other studies that have shown differences in morality between women and men are Franke, Crown, & Spake, 1997; Lippa, 1998; Muehlheusser, Roider, & Wallmeier, 2015.

Basing on the mentioned evidence, I would be more cautious than saying “…since no published work could be found on the topic of moral stereotypes and gender” (lines 152-154). I would rather say that the evidence on this issue is rare.

At the end of the introduction, the authors highlighted some limitations of the study. While I appreciate this part, I think they should just mention these limitations and discuss them more in depth (for instance, the last sentence, line 189-192) in the limitations section in the discussion (which is embedded in the text at the moment).

I would rename the section “Materials and Methods” as “Method”.

In the discussion, line 343, I cannot understand the sentence “…where people could large amounts of…”

I would avoid reporting statistics in the discussion (d values), could they be moved to the results section?

7. PLOS authors have the option to publish the peer review history of their article (what does this mean?). If published, this will include your full peer review and any attached files.

Reviewer #1: No

Reviewer #2: No

---

## [Author Response · Author response to Decision Letter 1]

18 Feb 2020

Rebuttal Letter

Comments

Reviewer #1: (No Response)

Reviewer #2: 

In the revised version, the authors have addressed the points raised by me and the other reviewer. In particular, I appreciated the adding of more considerations and references to the Pakistani context, which help understand and contextualize the findings.

I only have few concerns, which are outlined below.

With respect to gender differences in morality, the authors mostly reported studies on moral judgment. Even though I agree that there has been little research on morality in gender stereotypes, I suggest that the authors look at and eventually add the evidence of Leach, Carraro, Garcia, & Kang (2017) on stereotype of women as more trustworthy than men; Sheppard and Johnson (2019) on attractiveness and trustworthiness; Moscatelli, Menegatti et al. (2020, advanced online publication on Sex Roles) on the importance of morality for women in employment evaluation. Other studies that have shown differences in morality between women and men are Franke, Crown, & Spake, 1997; Lippa, 1998; Muehlheusser, Roider, & Wallmeier, 2015.

Added in introduction (Line 75- 98; 107-112) and References (Line 516-524; 532-536).

Basing on the mentioned evidence, I would be more cautious than saying “…since no published work could be found on the topic of moral stereotypes and gender” (lines 152-154). I would rather say that the evidence on this issue is rare.

Corrected Line 181-183.

At the end of the introduction, the authors highlighted some limitations of the study. While I appreciate this part, I think they should just mention these limitations and discuss them more in depth (for instance, the last sentence, line 189-192) in the limitations section in the discussion (which is embedded in the text at the moment).

Line 189-192 removed from introduction. Limitations and recommendations is made as a separate section (Line 448) and is discussed in detail (Line 449 - 454).

I would rename the section “Materials and Methods” as “Method”.

Corrected Line 219.

In the discussion, line 343, I cannot understand the sentence “…where people could large amounts of…”

Corrected Line 369.

I would avoid reporting statistics in the discussion (d values), could they be moved to the results section?

Removed from discussion section.

---

## [Editor Report · Decision Letter 2]

19 Feb 2020

Title: Accuracy of Consensual Stereotypes in Moral Foundations: A Gender Analysis

PONE-D-19-25204R2

Dear Dr. Inam,

We are pleased to inform you that your manuscript has been judged scientifically suitable for publication and will be formally accepted for publication once it complies with all outstanding technical requirements.

With kind regards,

Valerio Capraro

Academic Editor

PLOS ONE
---

## [Editor Report · Acceptance letter]

24 Feb 2020

PONE-D-19-25204R2 

 Accuracy of Consensual Stereotypes in Moral Foundations: A Gender Analysis

Dear Dr. Inam:

I am pleased to inform you that your manuscript has been deemed suitable for publication in PLOS ONE. Congratulations! Your manuscript is now with our production department. 

With kind regards,

on behalf of

Dr. Valerio Capraro 

Academic Editor

PLOS ONE